# Application of a text mining method in navigation and communication for enhancing maritime safety

**Paulina Hatłas-Sowińska**[1]*, **Leszek Misztal**[2]

1 Department of Mathematics Physics and Chemistry, Maritime University of Szczecin, Szczecin, Poland,
2 Faculty of Computer Science and Telecommunications, Maritime University of Szczecin, Szczecin, Poland

* p.hatlas-sowinska@pm.szczecin.pl

## Abstract

This paper introduces a model for the translation of natural language into ontology and vice versa in an autonomous navigation system of a sea-going vessel. The system comprehensively executes communication tasks at sea. The authors use machine learning methods in the field of text mining and basic and additional properties of ontologies. The newly developed ontology is applicable in shipping. The key elements of the prototype are the sequence of communication commands given from the ship's bridge, decomposition, extraction of the communication sequence and the rule base. The presented model has been implemented and verified in selected scenarios of collision situations at sea. The test results confirm that the assumptions, the designed system architecture and the algorithms in the prototype are correct.

## Introduction

The process of ship conduct requires continuous exchange and processing of navigational information. Whether the decisions made are correct depends on the extent, accuracy and reliability, as well as the appropriate perception of information. Ship navigators are required to use all available means to assess the navigational situation, including ship's equipment and systems, voice communication and other methods. Voice communication thus provides a channel of communication for obtaining additional information and, where appropriate, agreements. An analysis of maritime court decisions shows that, in cases of collisions, failure to establish voice communication with the other vessel was one of the allegations made against vessels involved in an accident. Wrong decisions can be caused by the failure to establish voice communication, improperly conducted communication or the misunderstanding of the information thus transmitted. Disadvantages of communicating orally include the problem of decoding a message on a semantic level, polarisation (the tendency to express extreme opinions), labelling (noticing problems by naming them rather than analysing them), mixing facts and conclusions, and static judgement (i.e. opinions are not verified even if elements of reality constantly change). The primary task of navigation is to ensure safety by avoiding hazards at sea. Direct communication established between ships followed by automated communication

**Data Availability Statement:** Hatłas-Sowińska Paulina, Misztal Leszek (2023). Data Set of ontology and communication scenario for method

of enhancing maritime safety. SEANOE. https://doi.org/10.17882/97041.

**Funding:** The authors received no specific funding for this work.

**Competing interests:** The authors have declared that no competing interests exist.

processes can minimize bad decisions and, consequently, wrong actions resulting in accidents at sea.

The majority of navigational accidents occurs due to human error. One classification of these errors was introduced by Reason [1]. Another publication [2] indicated the '80–20' rule, which states that up to 80% of accidents are due to human error and 20% are technical accidents.

This analysis of maritime accidents was made using the Maritime Statistical Yearbooks for the years: 2017–2022 [3] and the reports of the State Commission for the Investigation of Marine Accidents 2017–2022 [4].

The percentage of maritime accidents by type (Table 1) between 2017 and 2021 is comparable, i.e. the differences in the categories: serious accident, accident and incident range from 22% to 30%. In contrast, the difference between very serious accidents and the others is significant, ranging from 140% to 230%.

Table 2 illustrates the causes of marine accidents and incidents by year, with various factors responsible for the occurrence of these accidents: mechanical, human, external, other (e.g. organisational).

Table 2 shows that:

- In 2017, in 114 marine accidents and incidents, as many as 52 were influenced by the human factor. In seven cases the external factor was involved and in 52—the mechanical factor. Note that the human factor alone accounted for about 46%;

- In 2018, of 66 marine accidents and incidents, 26 were due to the human factor. In three cases there was an external factor and in 23 the mechanical factor was at play. This time the human factor accounted for about 40%;

- in 2019, of 73 marine accidents and incidents as many as 29 were influenced by the human factor. An external factor affected two cases, while 20 other cases were impacted by the mechanical factor. It was noted that the human factor alone accounted for about 39%;

**Table 1. Marine accidents by type.**

| Accident types | 2017 | 2018 | 2019 | 2020 | 2021 | SUMA |
|---|---|---|---|---|---|---|
| Very serious accident | 11 | 7 | 12 | 4 | 5 | 39 |
| Serious accident | 53 | 21 | 14 | 11 | 4 | 103 |
| Accident | 28 | 21 | 25 | 21 | 36 | 131 |
| Incident or event | 22 | 17 | 22 | 14 | 18 | 93 |
| TOTAL | 114 | 66 | 73 | 50 | 63 | |

Source: Data from the State Commission for the Investigation of Marine Accidents

**Table 2. Factors leading to marine accidents and incidents 2017–2020.**

| Factor | 2017 | 2018 | 2019 | 2020 | 2021 |
|---|---|---|---|---|---|
| human | 55 | 26 | 29 | 26 | 28 |
| mechanical | 52 | 23 | 20 | 15 | 17 |
| external | 7 | 3 | 2 | 3 | 2 |
| other | 3 | 14 | 22 | 6 | 16 |

source: [3]

- in 2020, of 50 marine accidents and incidents as many as 26 were caused by the human factor. Only three cases occurred due to the external factor, and 15 were due to the mechanical factor. It was noted that the human factor alone accounted for about 52%;

- In 2021, 63 marine accidents and incidents were recorded, of which 28 were caused by the human factor. In two cases the external factor was to blame, and 17 cases had mechanical causes. In this group the human factor was associated with about 45% of all accidents.

The trend for human factor to occur continues to be on the rise (even if the total number of accidents decreases). The research concept for solving the human error problem is the application of artificial intelligence algorithms and machine learning solutions from the field of text mining [5] also referred to as natural language processing [6]. This aims to automate communication between ships that have with an implemented autonomous navigation solution, as well as to automate communication between ships steered by the human navigator and a ship with an autonomous navigation system. The application of models for communication for the solution described in this work takes into account the two-way communication: messages sent by the human navigator to the autonomous system and vice versa. In the former case, voice commands are translated into natural language using speech recognition techniques [7]. Then they are subjected to text data processing and analysis techniques [8] related to tokenisation and word categorisation part-of-speech and command type recognition [9], sentence meaning analysis and message type classification [10] for the corresponding categories in the ontology. Finally, a command sequence expressed in the ontology is created, providing an unambiguous transfer of information to the autonomous system. In the case where the autonomous system initiates communication, the command sequence expressed in the ontology is subjected to the decomposition of the ontology sequence for one of the communication directions, which is then transformed according to the lexical rule base [11] into natural language. The transformation is based on the proposed mathematical model using ontology classes with a generating function and an interpreting function. In the final step, it is possible to utter a navigation-related message using a natural language produced by a speech synthesiser.

## Material and methods

### Analysis of the present state

One step towards increasing the level of safety is the introduction of innovative transport solutions. Efficiency of transport can be provided by improving transport processes through the use of appropriate IT tools to support these processes. Examples include e-navigation, monitoring and collision detection systems for road and rail traffic, road models for autonomous vehicles or ship systems and equipment. The latest methods in the field of communication are based on the introduction of intelligent conversational systems, i.e. computer programmes designed to simulate intelligent conversation through textual or verbal methods.

Today's maritime communication is based on convention requirements for ship equipment as well as personnel training that includes the knowledge of the International Convention for Preventing Collisions at Sea, often referred to as the Colllision Regulations, or COLREGs. Detailed communication procedures are set out in the International Radio Regulations issued by the International Telecommunication Union ITU). However, the process of conducting communications itself requires an extensive analysis of the situation based on existing procedures as well as the ship's equipment and systems.

One problem that arises in this context is that the communication process as understood in the field of maritime transport must be precisely defined. Its main task is to convey a message from the sender in such a way that it is unambiguously understood by the receiver. The large

amount of information, the diversity of its type and scope results in the need for processing, integration and selection, and most importantly, an unambiguous form and interpretation.

In radio communication, in addition to lexical and pronunciation problems, there are also technical issues, such as equipment failure. Numerous publications point to the communication system that is ineffective in certain situations. Communication errors are presented in [12].

There are current communication procedures recommended by the ITU, but they are complex and difficult for operators to follow.

In the analysis of the literature, particular attention was paid to these publications: [13–15]. Their main highlights include: failures in maritime communication implemented so far, indication that currently there is no ontology ready to be implemented in maritime communication, and presentation of benefits from the implementation of an ontology into the communication process.

A review of numerous publications in various scientific fields shows that the use of ontologies provides tangible benefits and translates into improvements in the quality of operation of a given system or industry. The current state of knowledge confirms that the research to date in the field of maritime transport shows a lack of an unambiguous and functioning ontology-based communication system. Applying ontologies and operating on semantic models extends capabilities of information gathering and processing.

To date, ontology has been used in the construction of several semantic models to describe the state and behaviour of ships at sea. Wen et al. proposed using ontology to build a semantic ship behaviour model (SMSB) based on a dynamic Bayesian network (DBN) to help represent and understand ship behaviour [16]. Van Hage et al. used ontology to build a Simple Event Model (SEM) to infer ship behaviour at different levels of abstraction, integrating knowledge from the network. That case is particularly interesting as it shows that ship position data is not sufficient for a navigator to fully understand the situation at sea [17]. Hagaseth et al. used existing semantic tools to extract the original meaning from maritime regulations texts to enhance the consistency of regulations, and to support compliance and enforcement by actors on ships and in ports [18].

The use of ontology-based information access techniques has also been used to enhance security and cyber security in ports [19]. Other applications of ontologies include underwater robots [20], decision support system in healthcare [21], ontology-based ship modelling [22], and ontology-based approaches to semantic sensors [23].

## Communication process

The specific character of maritime transport called for the construction of a separate, unique ontology allowing its application to automatic communication. Apparent deficiencies in verbal communication indicate the need to create a method that allows the inclusion of two parallel areas: navigation and communication. The elements contained in the navigation and communication ontologies are linked by models created using techniques from the field of text data mining for bidirectional communication between the human navigator and the autonomous navigation system. The way the models work and the steps used in them are described in the next section of the article. It also presents the translation between natural language and sequence in the communication ontology and vice versa. A prototype software with a graphical user interface for communication between an autonomous system and a human-operated system is also described. The prototype is based on data from the collision situation scenarios presented in the paper and performs the task of translation between the parties. The implemented prototype generates a natural language output, which can be used as input to a speech synthesiser library containing functions for voice generation. This makes it possible to imitate

human speech and convert the autonomous vessel communications to those understood by the traditional system (ship's personnel).

A method for building an ontology for maritime transport has been based on a model of the communication process, lexical analysis and the so-called loop method.

## Communication process model

The creation of a communication model, which forms the basis of the operations performed in the ontology, was based on the given literature, and on expert knowledge of maritime communication. There are many ready-made communication models, such as Lasswell's persuasive act model [24], the Shannon and Weaver model of signal transmission [25], Newcomb's triangular model [26], Schramm's community of experience model [27], the 'subcutaneous sting' model, the two-stage communication model, the agenda-setting process model of McCombs and Showa and others. None of the above-mentioned models is suitable enough to build a communication ontology for maritime transport on its basis.

The work of developing a customised communication process model was done in several steps: analysis of available communication models, identification of communication participants/ objects, definition of information content, definition of rules and techniques for communication, visualisation of the communication model. The created communication process model (Fig 1) contains the two-way communication activity as well as the ontology. A message sent from the sender to the receiver must pass through an ontology block: encoding, transmission and decoding. Encoding involves linking the type of the message to its content (function *f*), transmission delivers the message to function *g*, responsible for decoding the message and passing it to the receiver in a readable from. The reverse action, when the receiver intends to respond to the sender, i.e. feedback, is defined as the receiver's response to the message after decoding it.

**Lexical analysis.** In order to create a method for building an ontology for maritime transport, we need a vocabulary collection scheme for this domain (Fig 2). It consists in collecting data, then analysing it, selecting symbols for the word (if necessary) and implementation. The latter will show whether the concept is well placed in the overall ontology, or whether it should be changed or removed.

Lexical analysis is important in developing an ontology-based message structure, as this will allow data with a specific syntax to be read unambiguously. When loading such data, the

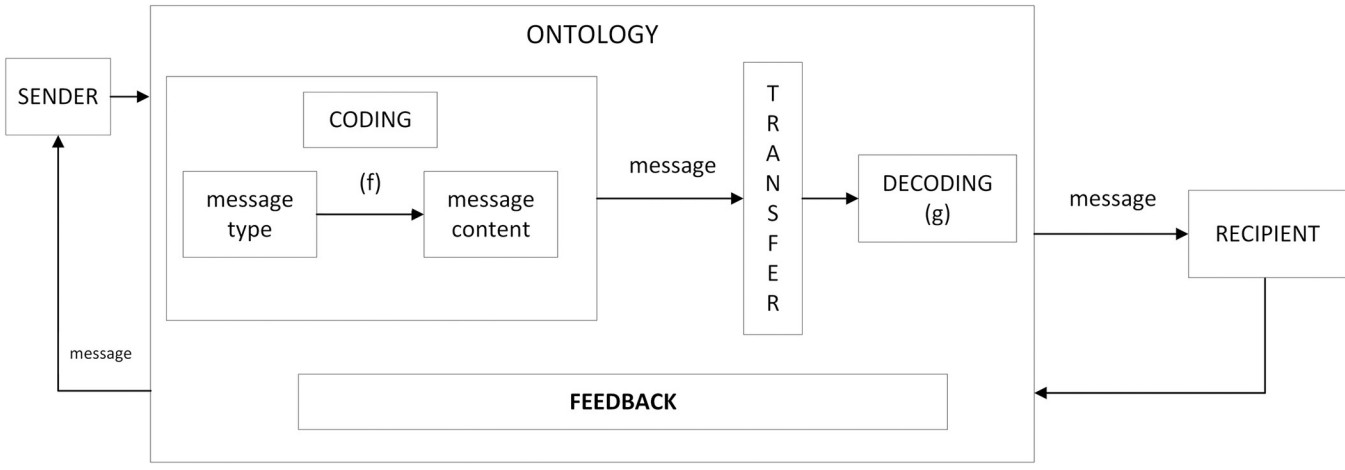

**Fig 1. Communication model.** Source: authors' work.

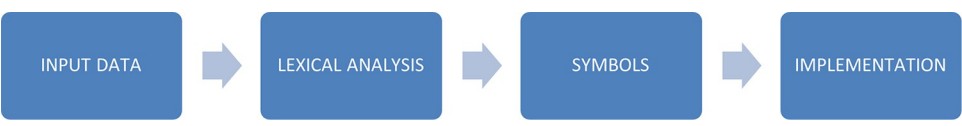

**Fig 2. Vocabulary collection scheme.** Source: authors' work.

syntax must be recognised before it can be processed. The coding action is to first divide the loaded string of words into smaller syntactic elements called lexemes, and only further analyse the string of lexemes. This constitutes a separate module dealing with this task, which aims to increase the efficiency of communication. During the lexical analysis of the ontology, a string of words is loaded and broken down into lexemes. However, what is passed on is not exactly lexemes. It is information representing the meaning of the lexeme. It is represented by a symbol and an optional attribute. The symbol represents information about the type of the lexeme. If the lexemes of a given type carry a certain "value", an attribute is attached to the symbol and is equal to this value.

**Loop method.** This method involves searching for a given word or pair of words in the order as contained in the ontology. Path finding in an ontology is based on two principles (Fig 3):

1. Calling up the addressee of the message, selecting the message type and category.

2. 2—An algorithm searches according to the order of ontology classes: selection of the first class, then subclasses, until the word from the message text is found;

checking for the word you are looking for in the main classes, if the word in question is not found in the body of the message, the search moves on to a subclass.

The loop terminates when the entire 'path' of the message has been obtained.

The method of building an ontology for maritime transport has been developed so that it ensures that the branch relating to a particular domain—in this case, the navigation ontology —can be continuously extended.

The present research work is based on transforming the ontological notation into natural language, using the text mining method.

## Theory/Calculation

### Ontology

By definition, ontology is a theory about any domain, describing concepts hierarchically in order to establish semantic relations. It is characterised by a logical theory that imposes constraints on logical models. Ontologies are created for various purposes [28]:

- to disseminate a common understanding of information structure among people or agent applications,

- to enable the knowledge in a particular field to be reused many times,

- to openly clarify assumptions about the chosen domain,

- to separate knowledge of the domain from the knowledge associated with operating the domain,

- to analyse knowledge of a specific domain.

The first definition was formulated by [29]: "an ontology is a formal, explicit specification of a shared conceptualisation".

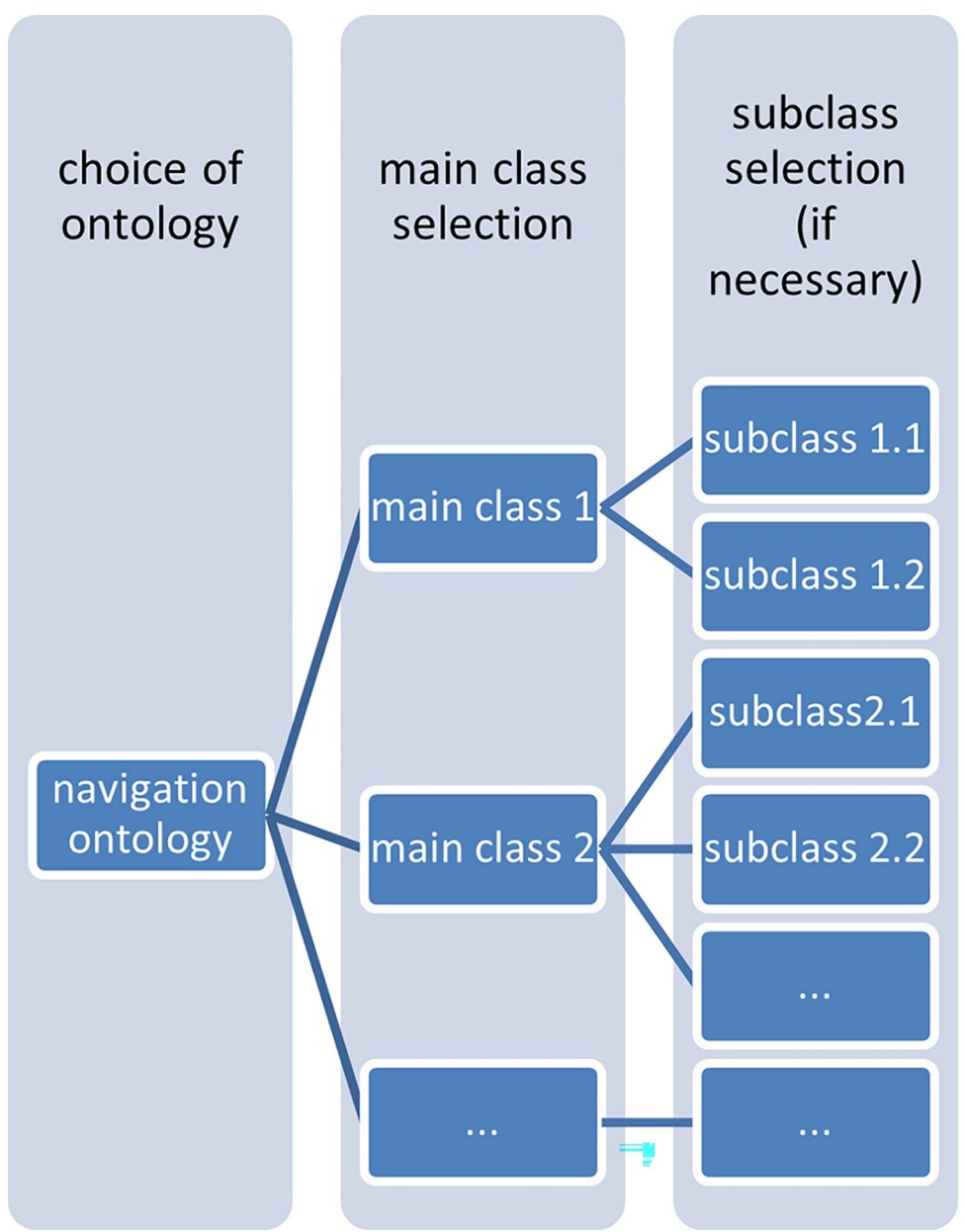

**Fig 3. Schematic of the loop method.** Source: authors' work.

An ontology can be written as a formula [30], where the set O defines the structure of concepts, the relations between them, as well as the theory about the model being defined.

For any broad domain, we can create different ontologies to describe the domain in different ways, so this ontology has been described using a new, proprietary, detailed formula to facilitate the collection of words for maritime transport.

$$O = \{A, K, R, f, g\} \tag{1}$$

where

O - ontology;

A - axiom of choice;

K - class of abstraction;

R - relationship

$f$ - message-generating function;

$g$ - message interpretation function.

The basic rules of each element in the set $O$ are given below [31–33].

**Axiom of choice.** One of the axioms of multiplicity theory stating that it is possible to construct a set (called a selector) containing exactly one element from each set belonging to a family of non-empty disjoint sets;

**Class abstraction.** If X is a non-empty set and $\approx$ is an equivalence relation on that set, then the sets $[x]$ for $x \epsilon X$ are called abstraction classes of relations $\approx$ w $X$. More precisely: the class $[x]$ is called the equivalence (abstraction) class of relations $\approx$ in $X$ determined by x or with representation of x if it satisfies the conditions:

- $[x] = \{y \in X : x \approx y\}$

- $\bigvee_{x,y \in X} (y \in [x] \Leftrightarrow (x \approx y)).$

The set of all equivalence classes of relations $\approx$ in $X$ is denoted as $X/_{\approx}$. For any elements: $x$, $x_1, x_2 \in X$, we have:

- $x \in [x]$

- $[x_1] = [x_2] \Leftrightarrow x_1 \approx x_2$

- $[x_1] \neq [x_2] \Rightarrow [x_1] \cap [x_2] = \emptyset.$

An equivalence relation $\approx$ defined on the set X, establishes the division of this set into non-empty and pairwise disjoint subsets, i.e. into classes of abstractions in this relation, in such a way that two elements $x, y \in X$ belong to the same class if and only if $x \approx y$.

**Relation.** Let there be given non-empty sets X and Y. A relation on the set $X \times Y$ is any subset of the Cartesian product of $X \times Y$. We shall denote the relation as $\rho$. If we consider a relation $\rho$ between elements of set X and elements of set Y, then $\rho \in X \times Y$,

We write then that for $x \epsilon X$, $y \epsilon Y$ the following occurs:

- $(x,y) \in \rho$ and we say that the couple $(x,y)$ belongs to a relationship $\rho$ or

- $x \rho y$ and then we say that the element x is in a relation $\rho$ with element y.

If X is an n-element set and Y is an m-element set, then there are $2^{n \cdot m}$ of all relations in the set $X \times Y$.

The relation $\rho \in X \times X$ is called an equivalence relation if it is reflexive ($\bigvee_{x \in X} x \rho x$), symmetric ($\bigvee_{x,y \in X} (x \rho y \Rightarrow y \rho x)$) and transitive $\bigvee_{x,y,z \in X} [(x \rho y \text{ and } y \rho z) \Rightarrow x \rho z]$. We will denote equivalence relations as: $\approx$:

- $x \approx y \Leftrightarrow x = y, \quad x, y \in R$

- $k \approx l \Leftrightarrow k \cdot l > 0, \quad k, l \in Z\{0\}$

- $x \approx y \Leftrightarrow x - y \, \epsilon \, Z, \quad x, y \in R$

The relation is reflexive because: $\bigvee_{x \in X} x - x = 0 \epsilon Z$

The relationship is symmetric because: $\bigvee_{x,y \in R} x - y \epsilon Z \Rightarrow y - x = -(x - y) \, \epsilon \, Z)$

The relationship is transitive because:

$$\bigvee_{x,y,z \in R} x - y \, \epsilon \, Z \; \textbf{\textit{i}} \; y - z \, \epsilon \, Z \Rightarrow x - z = (x - y) + (y - z)$$

The relation $\rho$ is a partial order relation if it is reflexive $\bigvee_{x \in X} x \rho x$, antisymmetric $\bigvee_{x,y,z \in X}[(x\rho y \; \textbf{\textit{and}} \; y\rho x) \Rightarrow x = y]$ and transitive $\bigvee_{x,y,z \in X}[(x\rho y \; \textbf{\textit{and}} \; y\rho z) \Rightarrow x\rho z]$.

The relation $\rho$ is a relation of linear order if it is reflexive $\bigvee_{x \in X} x\rho x$, antisymmetric $\bigvee_{x,y,z \in X}[(x\rho y \; \textbf{\textit{and}} \; y\rho x) \Rightarrow x = y]$, transitive $\bigvee_{x,y,z \in X}[(x\rho y \; \textbf{\textit{and}} \; y\rho z) \Rightarrow x\rho z]$, and consistent $\bigvee_{x,y \in X}(x\rho y \; \textbf{\textit{or}} \; y\rho x)$.

An example for a linear order relation: $x\rho y \Leftrightarrow x \leq y, \quad x, y \in R$

- reflexivity: $x \leq x$

- antisymmetry: $(x \leq y \text{ and } y \leq x) \Rightarrow x = y$

- transitivity: $(x \leq y \text{ and } y \leq z) \Rightarrow x \leq z$

- connectivity: $x \leq y \text{ or } y \leq x$

**Generating function *f* and interpreting function *g*.** The function f connects one element from the set Y with a selected element or elements from the set X, forming K, the so-called message body. The function g establishes the meaning of the information sent and the actions to be performed in relation to the message received. This function assigns a combination of elements from sets Y and X to the received message Ki.

The functions f and g are expressed by the formulae:

$$f : Y \times X \rightarrow K \quad f(y_n, X_k) = K_i \tag{2}$$

$$g : K \rightarrow Y \times X \quad g(K_i) = (y_n, X_k) \tag{3}$$

where

| | |
|---|---|
| $Y = \{y_1, y_2, \ldots, y_l\}$ | - a set of message types with associated category, ($l \in N$), |
| $y_n \in Y$ | - selected type with an associated message category, ($1 \leq n \leq l$) |
| $X = \{x_1, x_2, \ldots, x_m\}$ | - a set of navigational concepts (entities contained in the navigation ontology), ($m \in N$), |
| $X_k = \{x_{k1}, x_{k2}, \ldots, x_{kj}\} \subset X$ | - a set of entities in the k-th message, |
| $k$ | - the number of the transmitted message ($k \in N$), |
| $K$ | - set of messages, |
| $K_i$ | - $i$-th message from the set K, |
| $K_i = \{s_{in}, s_{i1}, s_{i2}, \ldots, s_{ij}\}$ | - individual words that appear in a message, ($i, j, n \in N$), |
| | where: |
| | $s_{in}$—n-th word in the i-th message $K_i$ from the set Y, |
| | $s_{ij}$—j-th word in the i-th message $K_i$ from the set X. |

The proof of the correct formulation of the above functions consists of three parts: part one covers the need to use the Cartesian product and formulates the properties of the function f, part two describes the set of values of the function f, part three presents the inverse function (continuous transformations).

There exists an ordered pair $(y,x)$ of two elements, assuming that: $y \in Y$ and $x \in X$. The Cartesian product of $Y \times X$ of the set $Y, X$ is the set of all ordered pairs $(y, x)$ such that:

$y \in Y$ and $x \in X$. Hence $Y \times X = \{(y, x) : y \in Y \wedge x \in X)$.

The Cartesian product was used to ensure that the words in the message were ordered.

Example:

The Cartesian product of the sets Y = {A_intention, T_information} and X = {course, position, passing} contains 6 ordered pairs:

$Y \times X$ = {(A_intention, course), (A_intention, position), (A_intention, passing), (T_information, course), (T_information, position), (T_information, passing)}.

The Cartesian product of the set Y cannot be formed with itself, i.e. $Y \times Y$, since the message K being constructed is of one type only. In contrast, one can form the Cartesian product of a set X with itself, i.e. $X \times X$ for the same word path, e.g. X = {course, alter course, to port, collision course} whereby, according to the definition given above, the ordered pairs are: $X \times X$ = {(course, alter course,), (course, collision course), (alter course, to port)}.

The proposed software research prototype uses the class sequences present in the ontology to execute communication at sea. After decomposing the entire ontology sequence into one-way communication, the prototype uses a proposed rule base that transforms the decomposed class sequences into sentences formulated in natural language. This goal is achieved by using the proposed mathematical definition of the Cartesian product for the communication and navigation ontology classes in the model and in the rule base.

The function f defined on non-empty sets Y, X such that:

$(Y \neq \emptyset \wedge X \neq \emptyset)$ with values in a non-empty set K ($K \neq \emptyset$), the relation: $f : Y \times X \rightarrow K$ satisfies the following conditions:

$$\bigwedge_{y \in Y} \wedge \bigwedge_{x_1, x_2 \in X} [(y, x_1) \in f \wedge (y, x_2) \in f] \Rightarrow y = y,$$

e.g: {T_information, course}—specify course and {T_information, position}- specify position;

the domain is $Y \times X = \{(y_n, X_k) : y_n \in Y \wedge X_k \subset X\}$ (i.e. the set of all elements belonging to the set X that are in a relation with at least one element from the set Y); the set of values of the function forms the set K.

**Text mining.** The increase of computer performance and the capacity of data storage systems has led to an extend in the use of artificial intelligence and machine learning algorithms [34] in science, administration, business, including transport applications [35]. One important methods are text mining algorithms [36], which use various types of data mining techniques, lexical analysis [37], rule induction methods [38], information extraction and other techniques [39]. In this way, it is possible to obtain new information, to understand the communicated intentions on the basis of information gathered from websites, books, opinions, various textual sources and information directly communicated by humans [40]. Practical natural language processing solutions [41] include, for example, machine translations from one language into another (for example, the commonly used Google Translator), understanding intentions expressed in natural language (e.g. car navigation systems), human dialogue systems (used in the ChatGPT conversation system), written text recognition systems (used in OCR applications), speech recognition systems used, for example, in intelligent home control systems, *a system* solutions for automatic analysis of the type of user feedback used by large sales networks.

The use of text mining algorithms based on ontologies enables the automation of maritime communication for the transmission of information between the captain on board and the vessel equipped with an autonomous control system. The automated prototype described herein is designed to avoid collisions and ensure safe navigation. The solution is implemented using an ontology, i.e. a set of hierarchical rules consisting of classes that, using information

sequence records concerning maritime communication and navigation, enable reliable and safe manoeuvres of ships at sea. Therefore, for two-way communication using a secure ontology-based protocol, two models have been proposed for the translation of information passed from an autonomous ship to a traditional ship. This is first model, which, using an ontology-based rule base, creates natural language sentences that can be communicated to the captain using a speech synthesiser. The other model aims to perform a natural language comprehension task (commands spoken on the navigational bridge) using a multi-step method including tokenisation, lexical analysis, part-of-speech recognition [9], command type identification, and, in the final step, generation of a class sequence expressed in the ontology based on the analysed sentences. The information transformed from natural language to a sequence of commands expressed in the ontology, executed in the described manner, carries out the transmission of maritime commands that are unambiguous and comprehensible to the autonomous system. A prototype application has been created for the first of the said models, which translates the sequence of classes in the ontology into natural language following the described steps of the model.

## Results

### Model of the two-way translation of natural language into/from ontology: Prototype

The proposed prototype of the ontology-based event/command sequence conversion model into natural language is presented in Fig 4 (below). The concept incorporates the conversion of command sequences from the navigation bridge exchanged with a ship with automatic communication into a natural language that will be understood by the navigator on a ship with traditional communication. The model carries out decomposition (when necessary) and extracts the communication sequence expressed in an ontology for one side of the communication. The next step is to create a natural language sentence using the rules defined for the sequence of events in the ontology. The mathematical method herein developed and presented is based on the Cartesian product using the generating function $F_g$ and the interpreting function $F_i$. The final step, not presented in the present model, will be the ability to utter the created sentence by one of speech synthesis solutions [42].

For the implementation of the described model, a prototype application written in the *Python* programming language [43] was created (Fig 5). It implements the individual steps of the model for the scenarios presented in the article, including the decomposition of event sequences in the ontology and the use of the *Rule_gen* class with programmed rules [40] to translate events in the ontology into natural language. Additionally, to visualise how the method works, the *Tkinter* library [44] was used to create a user-friendly prototype interface. The generated natural language sentence sequence based on the ontology sequence written in the scenario of Table 3 (Communication scenario (category: negotiation) with ontology notation for the relative bearing 112˚) is presented below.

The next prototype presents an implementation model for converting sentences from natural language to instruction sequences in an ontology.

The model consists of a number of steps, where the input are captain commands in the form of a sequence of natural language sentences, while the output is a sequence of commands in the form of an ontology (Fig 6). At the start of the operation, the model organises the data by tokenising it into individual words and sentences. It then discards words that are not relevant to the spoken commands, but are typical of human interactions (e.g. please, however, but, etc.). The next step is stemming, i.e. reducing the number of similar words that derive from a single parent (root). Lemmatisation, in turn, aims to reduce the number of words by using a

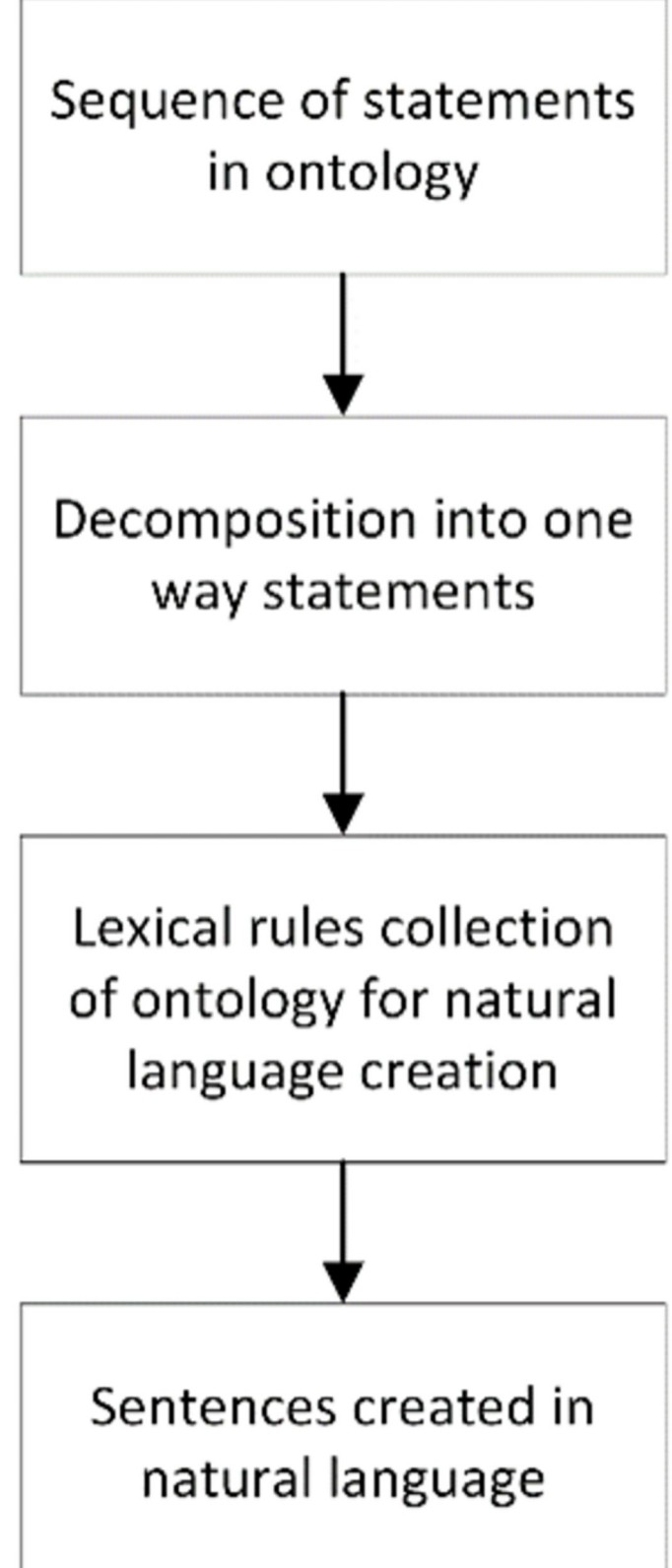

**Fig 4. Model for converting instruction sequences in an ontology into natural language.** Source: authors' work.

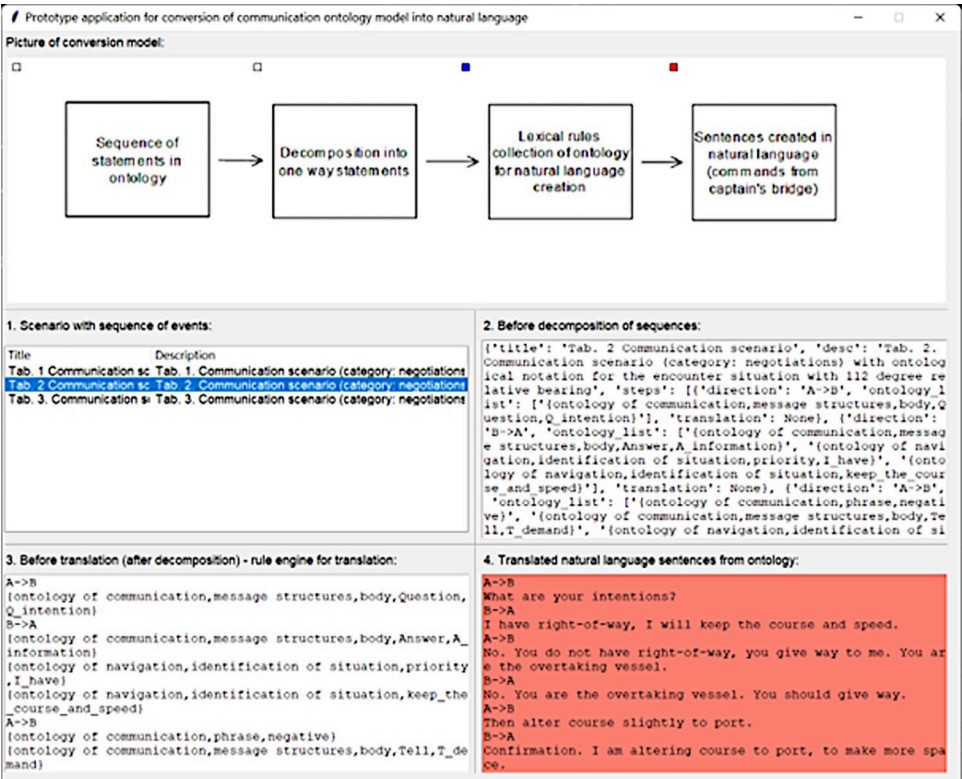

**Fig 5. Prototype operation for translating ontology event sequences into natural language.** Source: authors' work.

base word, thus reducing words with, for example, different endings. The information reduction steps are followed by the step of labelling the types of sentence parts such as nouns, verbs, adjectives and other, as well as identifying the types of commands. This is an important step before using the classifier to recognise the types of messages sent from the ship's bridge, such as information or request. Then, making use of the knowledge base in the form of rules, an operation will be carried out to create a sequence of commands in the form of classes from an ontology set. This form will be unambiguous and readable for the autonomous system.

The completed prototype communication model (Fig 7) has the following structure:

## Examples of collision situations and the model prototype

To achieve the objectives of this work, these authors employed the method of simulation tests conducted with the use of an ECDIS simulator. Some of the scenarios selected from previously prepared scenarios of ship encounters were executed. The ECDIS simulator consists of eight independent stations (ships) NaviTrainer 4000 from Transas cooperating with eight ECDIS NaviSailor 3000i stations. The simulator, installed at the Maritime University of Technology in Szczecin, makes it possible to simulate practically any ship encounter scenario in the selected shipping area, offering access to more than 10 ship models. Each ship model allows full use of the ship's equipment. The water area can be visually observed, and the navigator can also use any of the systems typically installed on the modern commercial ship bridge, including those used in ARPA and AIS scenarios. The ship models have full course and speed altering capabilities. The tests were conducted in the following configuration: two stations manned by trained navigators, who have full access to ship's equipment, with the ability to manoeuvre by altering

**Table 3. Communication scenario (category: Negotiations) with ontological notation for the encounter situation with 112 degree relative bearing.** Source: authors' work.

| Communication scenario: negotiation | Communication through ontologies | Natural language communication (text mining) |
|---|---|---|
| A to B: what are your intentions? | {ontology of communication, message structures, body, Question, Q_intention} | What are your intentions? |
| B to A: I have right-of-way, I will maintain course and speed | {ontology of communication, message structures, body, Answer, A_information},{ontology of navigation, identification of situation, priority, I_have}, {ontology of navigation, identification of situation, keep_the_course_and_speed}. | I have right-of-way, I will keep the course and speed. |
| A to B: no, you do not have right-of-way; you are the overtaking vessel; you give way to me | {ontology of communication, phrase, negative} {ontology of communication, message structures, body, Tell, T_demand}, {ontology of navigation, identification of situation, priority, I_have},{ontology of navigation, Identification of situation, Give way}, {ontology of communication, message structures, body, Tell, T_information}, {ontology of navigation, Identification of situation, Over taking}. | No. You do not have right-of-way, you give way to me. You are the overtaking vessel. |
| B to A: you are the overtaking vessel, you should give way | {ontology of communication, phrase, negative}, {ontology of communication, message structures, body, Answer, A_information}, {ontology of navigation, Identification of situation, Over taking}. {ontology of communication, message structures, body, Tell, T_expectation}, {ontology of navigation, Identification of situation, Give way}. | No. You are the overtaking vessel. You should give way. |
| A to B: then alter course slightly to port | {ontology of communication, message structures, body, Tell, T_information}, {ontology of navigation, Features information, Navigation information, course, Alter course, To port}. | Then alter course slightly to port. |
| B to A: OK, I am altering course to port to give you more space | {ontology of communication, roger}, {ontology of communication, message structures, body, Answer, A_information},{ontology of navigation, Features information, Navigation information, course, Alter course, To port}.{ontology of communication, message structures, body, Tell, T_information},{ontology of navigation, Features information, Ship maneuvering, more space}. | Confirmation. I am altering course to port to make more space. |

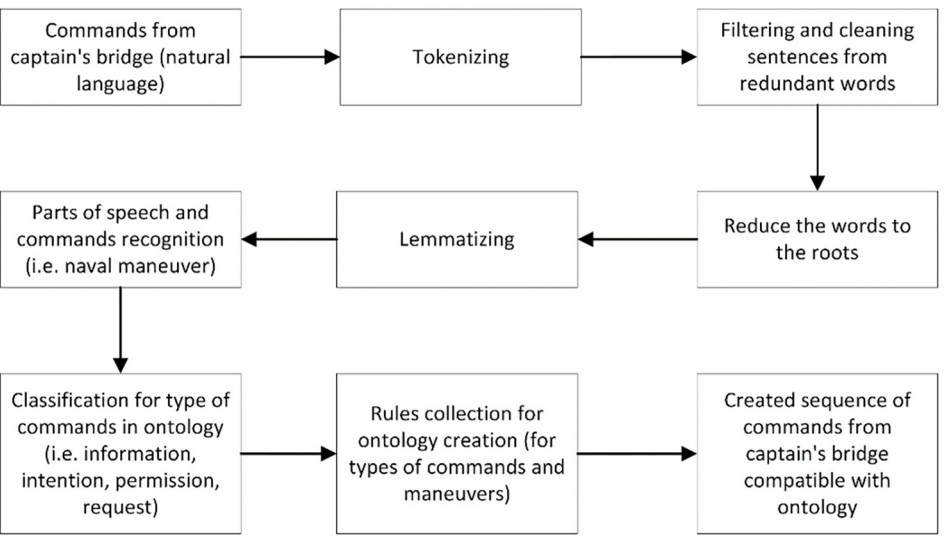

**Fig 6. A model for converting commands given on the navigational bridge from natural language to a sequence of commands in an ontology.** Source: authors' work.

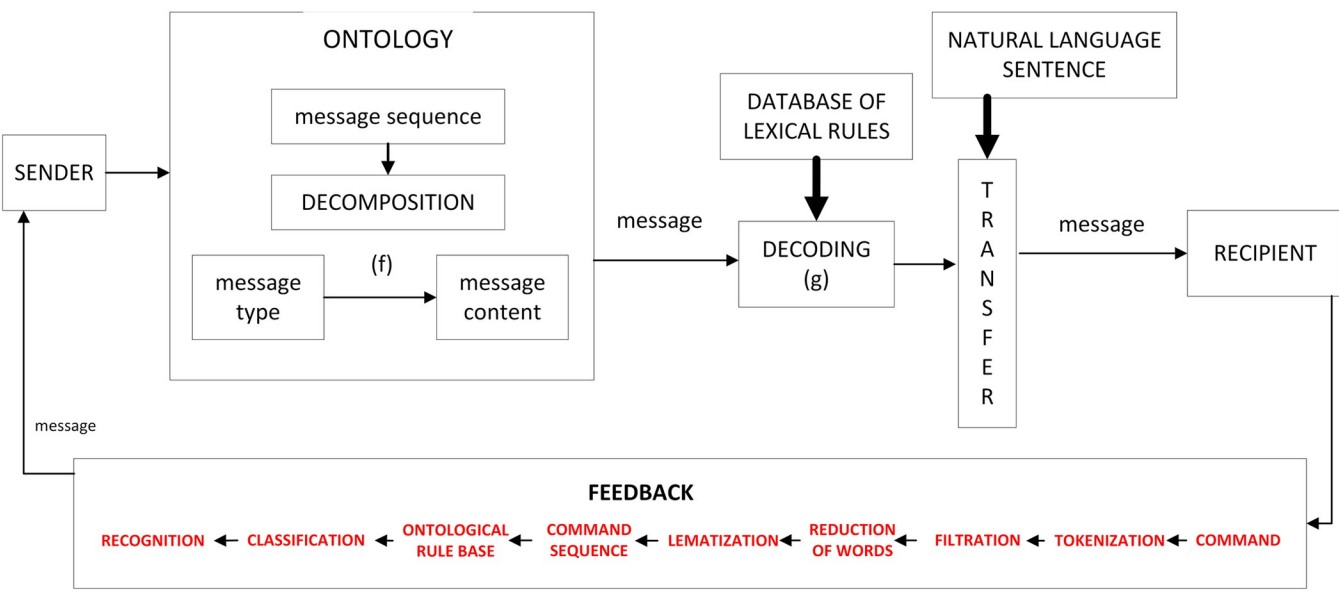

**Fig 7. Extended model of communication.** Source: authors' work.

course and speed, and are in charge of communication; each test participant sees a target ship in sight (visualisation), as well as by means of the ship's systems and equipment—radar, ARPA, AIS, ECDIS; at the start of the test, the relevant data is recorded from the AIS systems of two independent ships: "own" and "target".

Three encounter situations and two vessel models (non-autonomous, real-time) were selected for the study:

- crossing courses: ship A on course 270˚, and ship B on course 000˚; the meeting ships are of medium size (length ca. 170 m), both proceeding full ahead, in sight if each other;

- reciprocal or almost reciprocal courses, ship A on course 180˚, ship B on course 000˚; the meeting ships are of medium size (length ca. 170 m), both proceeding at full ahead and seeing each other;

- encounter with a vessel on relative bearing 112˚: vessel A proceeds on course 312˚, and vessel B on course 000˚; the meeting vessels are of medium size (length approx. 170 m), both proceeding full ahead and sea each other.

The selected scenarios are given as standard ship meeting situations as described in the COLREGs. Nevertheless, they raise many interpretative doubts about the safety of the manoeuvre being performed. The article presents a scenario of a collision situation of an encounter with relative bearing of 112˚, because similar situations are often misinterpreted, and one of two rules are taken as applicable: overtaking right-of-way or intersecting courses. Each situation was simulated several times (depending on how the situation developed). Each time, different communication exchanges and related manoeuvres by the ships were recorded. For these, appropriate scenarios were prepared and recorded, allowing the initial situation to be replayed several times with the possibility of any individual subsequent action to be taken by navigators.

Scenario: Ship B course: 000˚, ship A course: 312˚. Good visibility. Wind force: 2, sea state 1. Ship A sends an enquiry to B about intentions. Vessel B answers, sending information that it has the right of way and will maintain course and speed. Vessel A disagrees and sends a request

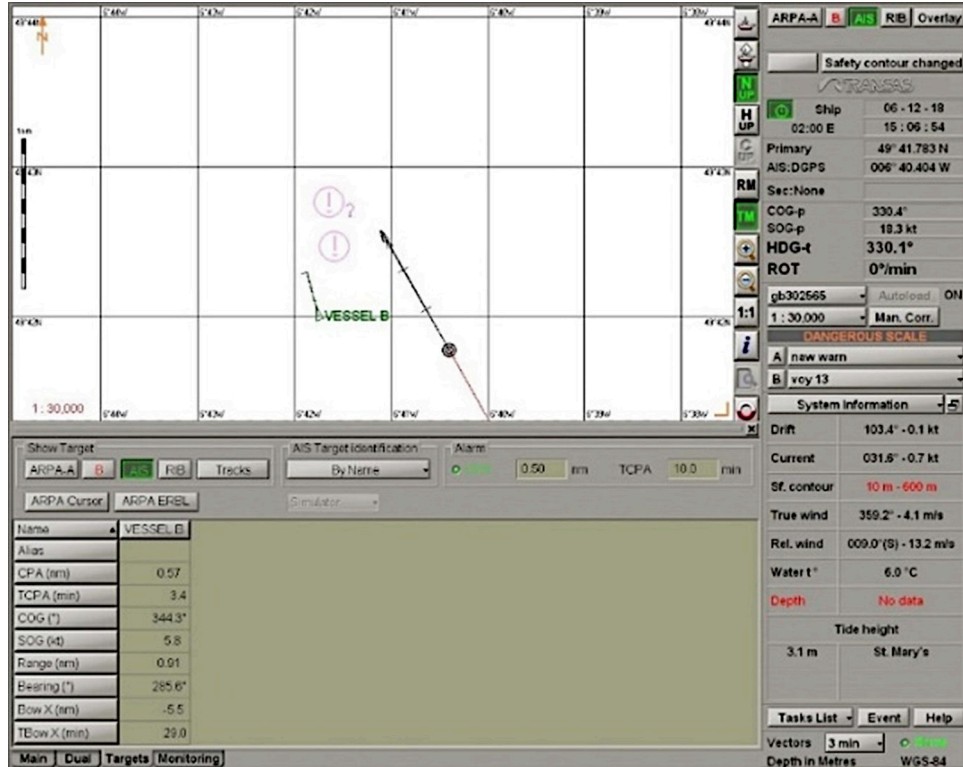

**Fig 8. Scenario: Relative bearing 112˚, negotiation.** Situation during a manoeuvre. Source: Transas ECDIS NaviSailor 3000i.

for ship B to give way. Vessel B also responds and requests ship A to give way, arguing that vessel A is the overtaking vessel. Vessel A informs vessel B that it alters course by 18˚ to starboard (to the right). Vessel B also alters course, but by 15˚ to port (to the left) to give more passage space ahead for vessel A (Fig 8). The vessels passed each other at a pre-fixed 0.6 Nm (Fig 9). The trajectory is shown in Fig 10.

## Discussion and future work

For each scenario, transformation of the communication using the ontology into natural language was carried out, based on the decomposition of the communication into individual commands in the ontology and constructing, according to the model, natural language sentences based on the rules contained in the model's rule base. The sentences built in this way can be successfully communicated to the navigator by generating the human voice using speech synthesis techniques and transmitting it using radio voice communication techniques available on board ships. The prototype software successfully implemented the translation for communication scenarios uploaded for testing.

In the future research work, efforts will be made to extend the prototype of automatic translation between the autonomous system and the human navigator based on the extension of the existing knowledge base of the model with further cases occurring in real scenarios that occur at sea. Another research step will be the cases of collision scenarios for communication in the other direction (feedback), where it will be necessary to realise a prototype based on the proposed model using text mining techniques to implement other scenarios actually occurring at sea. Besides, the authors intend to increase the accuracy of the proposed model for further

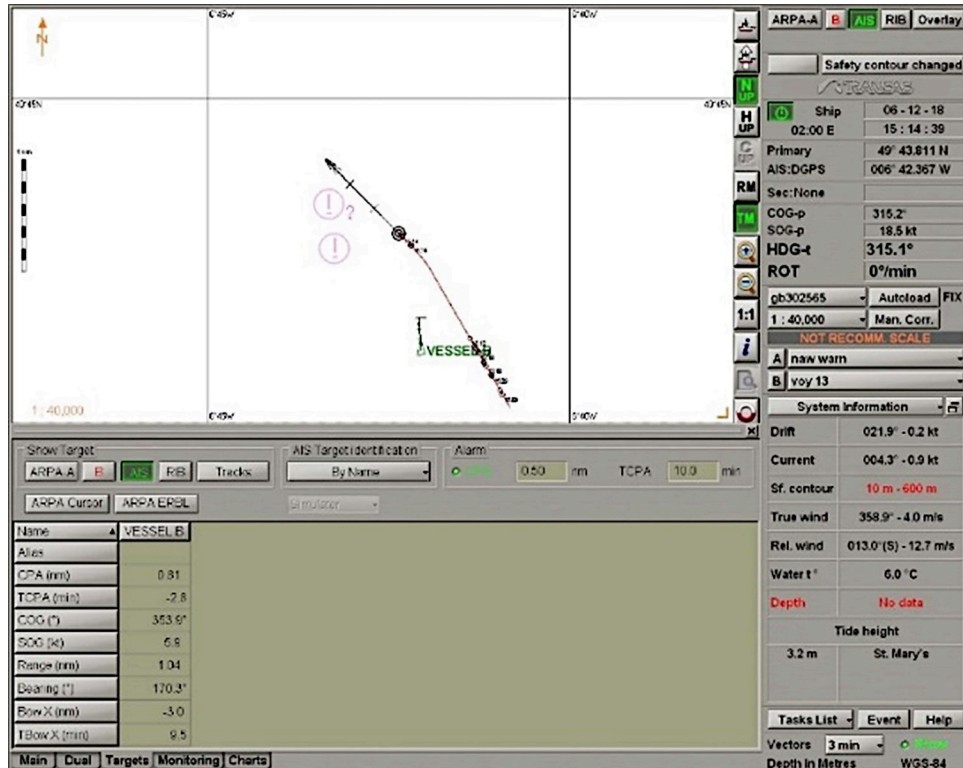

**Fig 9. Scenario: Relative bearing 112˚, negotiation.** Situation after the manoeuvre. Source: Transas ECDIS NaviSailor 3000i.

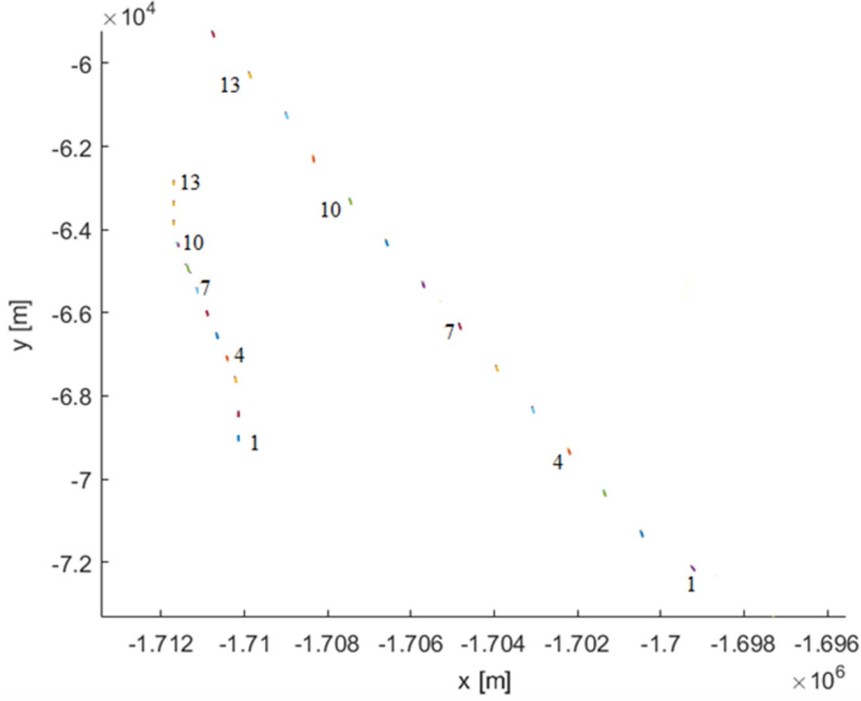

**Fig 10. Actual trajectories for the relative bearing 112˚ (scenario: Negotiation).** Source: authors' work.

natural language input used by ship captains. Ultimately, the work will be aimed to devise a fully functional two-way communication model that will enable accurate command recognition and the execution of the full scope of safe collision avoidance manoeuvres as they take place on waterways worldwide.

Proposed models for controlling commands from the captain's bridge to the autonomous system and automatically launched using empirical data from the presented test scenarios. Due to the operation of the first model—the article as a concept added to semi-autonomous maritime communication.

The target models will be tested based on a significant data set, where:

- for the first model, the input data will be a set of sequences of events in the ontology, while the output will be sentences in a natural language;

- for the second model, the input will be commands in a natural language, and the output will be a sequence of events in the ontology.

The test method will use developed models for each pair of elements from the set, where the input to the model will be input data, and the generated output data from the models will be compared with the collected output data (correct data). In this way, it will be possible to compare the correctness of the results and determine the accuracy coefficients of the method. Work on the proposed models is aimed at achieving the highest possible accuracy based on the collected data set.

## Conclusions

The automation of communication processes, including verbal communication between navigators on ships, may be one way to reduce collisions at sea.

This article presents assumptions, modes and forms of communication including automatic maritime communication systems as well as selected issues of the generation of outgoing messages, which are the result of automatically performed processes of analysis and interpretation of incoming messages. The models proposed for this purpose, based on text mining techniques using a rule base, made it possible to automatically translate natural language into ontology and vice versa, and consequently to carry out safe sea manoeuvres without the occurrence of collisions. These methods of automating communication processes, including verbal communication, are equally important for manned and unmanned, autonomous ships. The above also applies to other modes of transport.

The authors demonstrated the positive impact of using ontologies in the process of conducting ship-to-ship communication with a text mining approach. The article shows that the application of appropriate ontologies in a communication system could lead to the avoidance of collisions (or possibly to a significant reduction of their consequences). However, the application of the proposed solutions goes beyond collision or close-quarters situations. Sufficiently early communication in the indicated manner will allow potentially dangerous situations to be resolved much earlier, give navigators more time to analyse and observe how the situation develops, and reduce the stress associated with conducting last-minute manoeuvres.

Although correctly applied Collision Regulations and navigation procedures should be sufficient in the cases discussed, practice and the actual collision statistics show that the existing regulations are not always correctly interpreted. This is due to fatigue, haste, various pressures on navigating officers and, sometimes, inexperience and poor knowledge of the professional English language used in maritime navigation. In order to improve the automatic communication system and extend the way navigators communicate automatically, a model prototype was used for translating natural language into ontology (working both ways).

## Supporting information

**S1 Datasets.**
(ZIP)

## Author Contributions

**Conceptualization:** Paulina Hatłas-Sowińska, Leszek Misztal.

**Formal analysis:** Paulina Hatłas-Sowińska.

**Methodology:** Paulina Hatłas-Sowińska, Leszek Misztal.

**Software:** Leszek Misztal.

**Visualization:** Leszek Misztal.

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
