## [Decision Letter · Decision Letter 0]

24 Oct 2023

PONE-D-23-28679Application of a text mining methods in navigation and communication for enhancing maritime safety.PLOS ONE

Dear Dr. Hatłas-Sowińska,

Thank you for submitting your manuscript to PLOS ONE. After careful consideration, we feel that it has merit but does not fully meet PLOS ONE’s publication criteria as it currently stands. Therefore, we invite you to submit a revised version of the manuscript that addresses the points raised during the review process.

We look forward to receiving your revised manuscript.

Kind regards,

Sheraz Aslam

Academic Editor

PLOS ONE

Journal Requirements:

Additional Editor Comments:

Please consider the comments given by two reviewers in order to consider it for the next round.

Reviewers' comments:

Reviewer's Responses to Questions

**Comments to the Author**

1. Is the manuscript technically sound, and do the data support the conclusions?

Reviewer #1: Yes

Reviewer #2: Partly

2. Has the statistical analysis been performed appropriately and rigorously? 

Reviewer #1: Yes

Reviewer #2: No

3. Have the authors made all data underlying the findings in their manuscript fully available?

Reviewer #1: Yes

Reviewer #2: Yes

4. Is the manuscript presented in an intelligible fashion and written in standard English?

Reviewer #1: No

Reviewer #2: Yes

5. Review Comments to the Author

Reviewer #1: Dear Author(s),

Topic of the article is interesting. However, following comments should be addressed prior to further processing of the article.

1) Refer to whole article: Similarity is very high i.e. 32%

2) Refer to whole article: Too many formatting issues.

3) Refer to whole article: Authors have claimed that auto generated ontology will lead to safe sea communication. What do they think about machine error?

4) Refer to whole article: Language grammars are usually very complicated. Which language is targeted by the authors and how do they address grammar issue which may lead to communication of false signal?

5) Refer to Whole article: Scope of this study seems limited as the proposed model focuses on a single language. What about communication across different languages?

6) Refer to figure 2 & 3: Article is written in English however another language is used in figures.

7) Refer to sub-section 1.2.3: Authors need to include a flow chart for easier understanding of the process to the novice reader.

8) Refer to all figures: Figures quality is very low.

9) Refer to table 2: Content is replicated in table 2 and figure 5. Further, most of the references are very old. No reference from 2023. Authors need to include recent references.

Good luck.

Reviewer #2: The manuscript introduces an intriguing model for the translation of natural language into ontology and vice versa within the context of an autonomous navigation system for sea-going vessels. The authors aim to address the challenges associated with the oral communication of navigational information. The identified drawbacks of oral communication, including semantic decoding issues, polarization of extreme viewpoints, labeling problems, confusion between facts and conclusions, and static judgments, are well-highlighted.The topic is highly interesting and addresses a relevant issue in navigational communication.

Areas for Improvement:

1. The manuscript's formatting is subpar and does not adhere to standard conventions. Clear and organized presentation is crucial for effective communication of research findings.

2. Several important figures lack clarity, and their formats are not in accordance with the required standards. Clear and well-presented visuals are essential for readers to grasp the key concepts and findings.

3. The simulation validation process in the manuscript is deemed too simplistic. In real maritime communication scenarios, numerous pronunciation and technical issues are prevalent, yet these challenges are not adequately reflected in the simulated settings. A more comprehensive simulation that mimics the complexities of actual maritime communication should be considered.

The manuscript requires substantial revision, focusing on improving its formatting, enhancing the clarity of figures, and conducting a more realistic simulation validation. Addressing these issues will significantly enhance the overall quality of the manuscript.

6. PLOS authors have the option to publish the peer review history of their article (what does this mean?). If published, this will include your full peer review and any attached files.

Reviewer #1: **Yes: **Syed Muhammad Mohsin

Reviewer #2: No

---

## [Author Response · Author response to Decision Letter 0]

13 Nov 2023

Dear Reviewers,

 thank you very much for your thorough substantive assessment

and editorial of our article. Thank you for your positive opinions as well as the critical comments. They constitute important tips to improve the quality of our research work. Below we provide answers to the questions asked and responses to the comments included in the review.

1) Refer to whole article: Similarity is very high i.e. 32%

This article is a continuation of research on automatic communication in maritime transport. The authors continue their original research topic. There is little similarity between the activities in the studies. But this article introduces something new - the text mining method

2) Refer to whole article: Too many formatting issues. 

Thank you for this attention. The article has been corrected

3) Refer to whole article: Authors have claimed that auto generated ontology will lead to safe sea communication. What do they think about machine error? 

The phrase "machine error" can be understood as communication errors, user errors, calculation errors. The proposed communication model takes into account the occurrence of the mentioned errors. The authors state that machine errors may occur. Occurring in unusual, unpredictable and typical situations. These errors can come in two forms. The first one is – machine and computational errors. The second one is for processing errors: errors regarding data or numerical quantities.

With current research, the explanation for the above errors is non-existent. However, the authors do not deny that this is impossible.

4) Refer to whole article: Language grammars are usually very complicated. Which language is targeted by the authors and how do they address grammar issue which may lead to communication of false signal? 

The analysis of maritime case law indicates that in the event of a lack of connection between a voice call and a second call, this was one of the basic charges against the ships that took part in the occurrence. Decision errors may be caused by failure to initiate voice communication, its effects, or misunderstanding of the information conveyed in this way. These errors may be related to stress, which in turn affects the use of mental control and personal safety, self-assessment scores and situational awareness, disturbance of characteristic features, prolongation of decision duration. Errors may also be missing when it comes to using the English language. Disadvantages are determined in an oral form, including: problems with decoding the message at the semantic level, polarization (tendency to express extreme opinions), labeling (noticing problems by naming them, rather than analyzing them), mixing facts and occurrence as well as static assessment ( i.e. lack of verification of opinions regarding changes in elements of reality).

 The primary task of navigation is to ensure safe navigation by avoiding dangers during a sea voyage. The goal of establishing direct communication between ships and automating communication processes can reduce wrong decisions and, as a result, wrong actions resulting in maritime accidents. This mainly concerns dangerous situations requiring decisive action to avoid a collision, in particular excessive approach situations (a situation in which avoiding a collision requires concerted action by the navigators of the meeting ships).

Currently, communication at sea is based on conventional requirements regarding ship equipment and crew training. At the level of the international convention, rules for the law of the sea route (COLREGs) have also been established. Detailed communication procedures are specified in the International Radio Regulations issued by the ITU (International Telecommunication Union). However, the communication process itself requires a broad analysis of the situation based on existing procedures, as well as ship equipment and systems.

 The MPDM regulations are designed to safely conduct maneuvers, e.g. passing, overtaking, and especially anti-collision maneuvers - without the use of voice and/or radio-electronic communication. Maneuvering and warning signals (light and sound) are permitted for ships that are mutually visible. However, ambiguities and discrepancies in the interpretation of terms such as "safe distance", "early enough", "change course to the right" or the interpretation of the meeting situation (e.g. overtaking or crossing courses), introduced the practice of correspondence, most often by phone, between navigators meeting ships.

The communication process is divided into individual elements: the sender, i.e. the initiator, the recipient and the message. The person initiating communication performs coding, i.e. transforming thoughts into words that he thinks will be understandable to the recipient, thus creating a message. Then comes the communication channel, i.e. the way in which the content is to be conveyed. The message reaches the said recipient. Decoding, i.e. the interpretation of the received information, depends on its perceptual capabilities.

 In such a situation, a problem arises - the communication process understood in the field of maritime transport must be precisely defined. Its main task is to convey the message from the sender in such a way that it is clearly understood by the recipient. A large amount of information and the diversity of its types and scopes result in the need to process, integrate and select it. And most importantly – a clear form and interpretation. An example of ambiguity that a navigator may understand in different ways is: heave up the line/send heaving line.

The article presents sample dialogues in scenarios and the recording of messages in ontological notation - in English. The reason is that English is considered the basic language in maritime navigation.

There are current communication system procedures recommended by the ITU (International Telecommunications Union), but they are complicated and difficult to assimilate by operators.

The current state of knowledge confirms that all previous research in the field of maritime transport does not indicate a clear and working communication system based on ontology.

The use of ontology and the beginning of operating on semantic models contributes to obtaining new possibilities related to the collection and processing of information.

5) Refer to Whole article: Scope of this study seems limited as the proposed model focuses on a single language. What about communication across different languages?

Due to maritime law, which requires communication in English, the authors deal only with this language.

6) Refer to figure 2 & 3: Article is written in English however another language is used in figures. 

Thank you for this attention. The drawings have been corrected

7) Refer to sub-section 1.2.3: Authors need to include a flow chart for easier understanding of the process to the novice reader. 

Thank you for your attention. We added information about the process in the loop method

8) Refer to all figures: Figures quality is very low. 

Thank you for your attention. The drawings have been corrected

9) Refer to table 2: Content is replicated in table 2 and figure 5. Further, most of the references are very old. No reference from 2023. Authors need to include recent references. 

Thank you for your attention. The drawings and tabels have been corrected

The authors use the Statistical Yearbook of Maritime Economy 2022.

The provided data was included in the article.

In January 20234 - data from 2022 will be available.

This chapter contains information on accidents at sea to ships of Polish nationality, and marine salvage. 2. Data on accidents at sea cover ships of either Polish or non-Polish nationality when the incident or accident at sea occurred in Polish internal waters or Polish territorial sea. Passenger ro-ro ferries or highspeed passenger ships are also included when the incident or accident occurred outside the internal waters or territorial sea of the EU member state provided that the last port of call of that ship was a Polish Republic's seaport. In addition these statistics cover ships of gross tonnage below 50, i.e. fishing boats, yachts or tugs

---

## [Decision Letter · Decision Letter 1]

5 Jan 2024

PONE-D-23-28679R1Zastosowanie metod eksploracji tekstu w nawigacji i łączności dla poprawy bezpieczeństwa morskiego.PLOS ONE

Dear Dr. Hatłas-Sowińska,

Thank you for submitting your manuscript to PLOS ONE. After careful consideration, we feel that it has merit but does not fully meet PLOS ONE’s publication criteria as it currently stands. Therefore, we invite you to submit a revised version of the manuscript that addresses the points raised during the review process.

Please revise empirical study as suggested by Reviewer 2 and resubmit the revised copy.

We look forward to receiving your revised manuscript.

Kind regards,

Sheraz Aslam

Academic Editor

PLOS ONE

Additional Editor Comments (if provided):

Since Reviewer 2 is not satisfied related to empirical study,

"I still have reservations regarding the empirical validation of the method of this study".

Please consider empirical validation again.

Reviewers' comments:

Reviewer's Responses to Questions

**Comments to the Author**

1. If the authors have adequately addressed your comments raised in a previous round of review and you feel that this manuscript is now acceptable for publication, you may indicate that here to bypass the “Comments to the Author” section, enter your conflict of interest statement in the “Confidential to Editor” section, and submit your "Accept" recommendation.

Reviewer #1: All comments have been addressed

Reviewer #2: All comments have been addressed

2. Is the manuscript technically sound, and do the data support the conclusions?

Reviewer #1: Yes

Reviewer #2: Partly

3. Has the statistical analysis been performed appropriately and rigorously? 

Reviewer #1: Yes

Reviewer #2: I Don't Know

4. Have the authors made all data underlying the findings in their manuscript fully available?

Reviewer #1: Yes

Reviewer #2: Yes

5. Is the manuscript presented in an intelligible fashion and written in standard English?

Reviewer #1: Yes

Reviewer #2: Yes

6. Review Comments to the Author

Reviewer #1: Dear Author(s),

My comments are satisfactorily addressed in the revised version. I have no more comments.

Good luck.

Reviewer #2: Although the revised manuscript has addressed most of the questions raised in the initial review and made appropriate adjustments to the text. However, I still have reservations regarding the empirical validation of the method of this study. While authors have validated your method in three specific encounter situations, I believe that this may not sufficiently demonstrate its applicability and effectiveness across a broader range of real-world scenarios.

7. PLOS authors have the option to publish the peer review history of their article (what does this mean?). If published, this will include your full peer review and any attached files.

Reviewer #1: **Yes: **Syed Muhammad Mohsin

Reviewer #2: No

---

## [Author Response · Author response to Decision Letter 1]

26 Jan 2024

Dear Reviewer,

 Thank you very much for your further comments to our article. To ensure the quality of our research work is as good as possible, we provide the answer to the question below.

Data validation is a solution that uses research with the efficiency of empirical data. We try to ensure that the data of the congregations in our research are thorough and detailed. Validation is important because if it is false or incomplete, data may be exposed and incorrect decisions may be made.

Proposed models for controlling commands from the captain's bridge to the autonomous system and automatically launched using empirical data from the presented test scenarios. Due to the operation of the first model - the article as a concept added to semi-autonomous maritime communication.

The target models will be tested based on a significant data set, where:

- for the first model, the input data will be a set of sequences of events in the ontology, while the output will be sentences in a natural language;

- for the second model, the input will be commands in a natural language, and the output will be a sequence of events in the ontology.

The test method will use developed models for each pair of elements from the set, where the input to the model will be input data, and the generated output data from the models will be compared with the collected output data (correct data). In this way, it will be possible to compare the correctness of the results and determine the accuracy coefficients of the method. Work on the proposed models is aimed at achieving the highest possible accuracy based on the collected data set.

Data analysis software is an indispensable tool in empirical research. The authors plan to conduct advanced statistical analyses, which will enable a better understanding of the collected data and drawing accurate conclusions. This is the direction of further research work by the authors.

---

## [Editor Report · Decision Letter 2]

13 Feb 2024

Application of a text mining methods in navigation and communication for enhancing maritime safety

PONE-D-23-28679R2

Dear Dr. Hatłas-Sowińska,

We’re pleased to inform you that your manuscript has been judged scientifically suitable for publication and will be formally accepted for publication once it meets all outstanding technical requirements.

Kind regards,

Sheraz Aslam

Academic Editor

PLOS ONE
---

## [Editor Report · Acceptance letter]

11 Mar 2024

PONE-D-23-28679R2 

PLOS ONE

Dear Dr. Hatłas-Sowińska, 

I'm pleased to inform you that your manuscript has been deemed suitable for publication in PLOS ONE. Congratulations! Your manuscript is now being handed over to our production team.

Kind regards, 

on behalf of

Dr. Sheraz Aslam 

Academic Editor

PLOS ONE